# OpenReview forum: "Reinforcement Learning with Missing Context to Mitigate Reward Hacking from Training Only on Golden Answers"
_ICLR.cc/2026/Conference — ICLR 2026 Conference Withdrawn Submission_

### Official Review · Reviewer_owYq · 2025-10-28

**Soundness:** 2
**Presentation:** 2
**Contribution:** 2
**Rating:** 2
**Confidence:** 3

**Summary:**

The paper propose a new framework and benchmark to evaluate and improve the abiltiy of answer unclear questions (missing context).

**Strengths:**

- RLMC significantly improves robustness, reduces hallucinations, and enables models to recognize uncertainty and seek clarification when context is missing.

- if opensource, the benchmark will help researcher explore this direction.

**Weaknesses:**

- The code repo is empty.

- The general reasoning ability will decrease after RLMC. There is clear trade-off between HRB and final performance.

**Questions:**

- What will be the performance if introduce a small finetuned classification model to decided whether the question is ideal.

- How is the proportional division in the section “A Partition of the Trajectory Space” determined?

- It will be better to add a comparsion to RLVR/SFT only methods.

---

> ### Author Response · Authors · 2025-11-17
> **Official reply to Reviewer owYq by Authors**
>
> Dear Reviewer owYq,
>
> Thank you very much for your thorough review and valuable feedback on our paper. We are pleased that you recognize RLMC's significant improvements in model robustness, hallucination reduction, and its ability to identify uncertainty and seek clarification in the presence of missing context. We also appreciate your sentiment that our benchmark, if open-sourced, would greatly contribute to advancing research in this direction. We would like to provide detailed clarifications and explanations regarding your specific concerns.
>
> ---
> > Regarding the concern about comparisons with RLVR/SFT-only methods
>
> We appreciate your focus on baseline comparisons. However, our paper **already includes the comparisons you mentioned with RLVR/SFT-only methods**. Please refer to **Table 1 (Page 8)**:
>
> *   **Cold-start SFT** is precisely our **SFT-only** baseline, representing the model without any RL fine-tuning.
>
> *   **Vanilla PPO** represents the standard **verifier-based RL** approach, which is what you refer to as **RLVR**. This baseline is trained solely on answerable, well-specified problems and rewards correct final answers, thereby embodying the "reward hacking" paradigm criticized in our paper.
>
>
> Therefore, our experimental design comprehensively covers these important baselines, demonstrating RLMC's superiority on this foundation.
>
> ---
> > Regarding the concern about the trade-off between general reasoning ability and HRB performance
>
> We acknowledge that a trade-off between specific capabilities and general abilities is a critical and common challenge in model training **\[1, 2\]**. However, our paper **has already extensively discussed this point**: a core contribution of RLMC is precisely its ability to achieve a **near-optimal balance** in this trade-off. As shown in **Table 1 (Page 8)**:
>
> *   The **IDK-RL** method suffers **a significant performance drop** on general reasoning benchmarks (e.g., GSM8K decreased by 5.83 percentage points), which we term the "hallucination tax."
>
> *   In contrast, **RLMC** performs **comparably** to the highly specialized **Vanilla PPO** baseline on these general reasoning tasks, with **only minor drops** (e.g., GSM8K decreased by 2.35 percentage points). Concurrently, RLMC achieves the highest score across all strategies on HRB, reaching 51.73.
>
>
> This clearly demonstrates that RLMC successfully strikes a balance between **substantially improving robustness** (addressing reward hacking) and **maintaining strong general reasoning capabilities**. This avoids the pitfall of traditional methods that significantly sacrifice general performance to reduce hallucinations. By guiding the model toward advanced hypothetical reasoning, rather than simple abstention, we achieve this verifiable and effective trade-off.
>
> ---
> > Regarding the concern about the feasibility of introducing a small classification model to determine if a question is "ideal"
>
> We appreciate this insightful suggestion. However, relying  solely on a classification model to identify the state of a question **cannot address the core challenge of** **generative hypothetical reasoning** that our RLMC framework targets. A classification model can only output category labels, but it **cannot**: explain _why_ a question is not ideal, generate assumption-based conditional solutions, or proactively initiate clarifying dialogues—all of which require complex generative capabilities and a deep understanding of problem structure.
>
> Our RLMC reward system is a **finely designed multi-level value function**. It not only rewards "uncertainty identification" (0.3 points) but explicitly **encourages and cultivates** the model to go beyond simple abstention towards advanced, generative hypothetical reasoning by assigning **higher reward values** (0.6 points for Conditional Formulation, 1.0 point for Active Elicitation). Our **Generative Reward Model (GRM)**, utilizing a large model (Qwen3-235B-A22B-Instruct-2507), is specifically designed to understand and score these complex reasoning trajectories, rather than merely identifying problem types. Therefore, while classification might be useful in certain scenarios, for cultivating LLMs' complex, uncertainty-aware reasoning abilities, it is an oversimplified approach.

---

> ### Author Response · Authors · 2025-11-17
> **Official reply to Reviewer owYq by Authors （Part 2）**
>
> ---
> > Regarding the concern about "proportional division" in the section “A Partition of the Trajectory Space”
>
> We are not entirely clear about **what "proportional division" in your phrasing refers to** within our "A Partition of the Trajectory Space" section on Page 6. We would first like to clarify that this section discusses **not the proportion of training data**, but rather our classification of different reasoning behaviors a model might exhibit when facing missing-context problems, and the reward values assigned to these behaviors. We surmise you might be asking _why_ we partition model responses into these five categories (SH, EA, Abs, Cond, Elicit).
>
> We chose these five behavioral categories based on inspirations from cognitive science principles of **hypothetical reasoning, human problem-solving under uncertainty, and metacognition** **\[3, 4, 5, 6\]**. This five-tier classification **originates from** the progressive cognitive maturity in human decision-making and problem-solving under uncertainty, finding solid theoretical grounding in cognitive science:
>
> *   **Silent Hallucination (SH)** represents a **lack of metacognitive monitoring \[6\]**, where the agent fails to recognize its knowledge deficits, leading to confidently incorrect answers.
>
> *   **Explicit Assumption (EA)** and **Abstention (Abs)** signify the **initial establishment of metacognitive monitoring \[6\]**, where the agent acknowledges missing information or uncertainty, choosing to transparently state assumptions or admit insolvability.
>
> *   **Conditional Formulation (Cond)** reflects the ability, under incomplete but structured information, to derive **partial or conditional solutions** through **flexible problem representation \[5\]** – a key strategy for reasoning under limited information.
>
> *   **Active Elicitation (Elicit)** represents the **highest level of metacognitive control \[6\] and proactive problem-solving strategy \[5\]**: when critical information is identified as missing (metacognitive monitoring), the agent proactively acts (metacognitive control) to acquire it, thereby overcoming uncertainty and advancing problem-solving. This is a **crucial step in seeking satisfactory solutions under bounded rationality.**
>
>
> This comprehensive progression illustrates the maturity ladder for an intelligent agent navigating complex, uncertain environments. This design guides the model's policy to shift from low-value hallucinatory behaviors towards high-value proactive, uncertainty-aware reasoning. We have empirically validated the effectiveness of this reward system through experimental results (e.g., **Table 1 (Page 8)** and **Figure 4b (Page 9)**).
>
> If your phrasing of "proportional division" refers to the proportion of missing-context problems in the training data, this aspect is discussed in detail in **Section 4.3 "In-depth Analysis" (Page 7)** and **Table 2a (Page 8)**, where we conclude that a 30% ratio of missing-context training data is optimal.
>
> ---
>
> Finally, regarding your concern about the status of the code repository, we are pleased to inform you that the code, data, and HRB benchmark are now fully open-sourced at the anonymous link: [https://anonymous.4open.science/r/RLMC-RBG](https://anonymous.4open.science/r/RLMC-RBG)
>
> We hope these detailed explanations fully address your concerns.
>
> Sincerely, Anonymous Authors
>
> ---
>
> **References:**
>
> \[1\] Yang R, Pan X, Luo F, et al. Rewards-in-context: multi-objective alignment of foundation models with dynamic preference adjustment\[C\]//Proceedings of the 41st International Conference on Machine Learning. 2024: 56276-56297.
>
> \[2\] Guo Y, Cui G, Yuan L, et al. Controllable preference optimization: Toward controllable multi-objective alignment\[J\]. arXiv preprint arXiv:2402.19085, 2024.
>
> \[3\] Kuhn D. Children and adults as intuitive scientists\[J\]. Psychological review, 1989, 96(4): 674.
>
> \[4\] Pearl J. Causality\[M\]. Cambridge university press, 2009.
>
> \[5\]Simon H A, Dantzig G B, Hogarth R, et al. Decision making and problem solving\[J\]. Interfaces, 1987, 17(5): 11-31.
>
> \[6\]Metacognition: Knowing about knowing\[M\]. MIT press, 1994.

---

> ### Comment · Reviewer_owYq · 2025-11-20
>
> Thank you for your response. For Regarding the concern about "proportional division" in the section “A Partition of the Trajectory Space”, what I want to ask is What are the ratios here, for example 1:1:1:1:1 or something else? And how is this related to the value that you ultimately assign manually?
>
> For the PPO baseline, it seems not very fair to compare your methods with PPO which only see the answerable dataset, since your method have seen more data? Why not compare ppo with whole dataset   vs   RLMC with ( whole dataset + part of the dataset with Perturbation). The key thing is it is not fairable PPO only saw 600 correct data while RLMC saw 600correct data + 400 incorrect data.
>
> Because of other answers, I decide to improve the score to 4 but lower the confidence. I will reconsider if these two questions are answered.

---

> > ### Author Response · Authors · 2025-11-20
> > **Official reply to Reviewer owYq by Authors**
> >
> > Dear Reviewer owYq,
> >
> > Thank you for your follow-up questions, which help us further clarify our methodology and experimental design.
> >
> > > Regarding the concern about "proportional division" in the section “A Partition of the Trajectory Space”, what I want to ask is What are the ratios here, for example 1:1:1:1:1 or something else? And how is this related to the value that you ultimately assign manually?
> >
> > Thank you for explicitly asking about ratios. We understand you might be confused about the connection between these concepts. To provide you with a clearer understanding, we clarify the nature of "A Partition of the Trajectory Space" and its relationship with reward values:
> >
> > **Clarification on "A Partition of the Trajectory Space":**
> >
> > The term "partition" in Section 3.2 refers to a **conceptual taxonomy of model behaviors**, not a proportional division of data or trajectories. Specifically, **no ratios are involved**. The five behavioral categories (Silent Hallucination, Explicit Assumption, Abstention, Conditional Formulation, Active Elicitation) are disjoint sets that conceptually cover the entire space of possible model responses to underspecified problems. There is no 1:1:1:1:1 or any other pre-defined ratio that we aim for or that exists within our framework. RLMC is a data synthesis and curation framework, not a data recipe that dictates behavioral proportions.
> >
> > **The evaluation and reward assignment process is as follows:**
> >
> > 1.  **Model generates response:** Our model (policy $\pi\_\theta$) generates a response $\tau$ to a query $s'$.
> >
> > 2.  **GRM classifies response:** Our Generative Reward Model (GRM) classifies this response $\tau$ into one of the five pre-defined behavioral categories (e.g., SH, EA, Abs, Cond, Elicit). This classification is the application of "A Partition of the Trajectory Space."
> >
> > 3.  **Reward assigned:** Based on this classification, a corresponding **manually set reward value** is assigned (e.g., $1.0$ for Elicit, $-1.0$ for SH).
> >
> > 4.  **Model learns:** The model's objective is to maximize its expected reward. It adjusts its parameters to generate high-reward behaviors more frequently and low-reward behaviors less frequently.
> >
> >
> > **Relationship between "Partition" and "Manually Assigned Reward Values":**
> >
> > The manually assigned reward values serve as the **direct signal driving the model's learning**. The model naturally adjusts its behavior according to these scores. Therefore, the final proportion of various behaviors generated by the model is an **emergent outcome of the model learning to achieve higher total rewards**, rather than being pre-imposed by us. We use these preference-based reward values to guide the model towards more intelligent and responsible hypothetical reasoning behaviors.
> >
> > ---

---

> > ### Author Response · Authors · 2025-11-20
> > **Official reply to Reviewer owYq by Authors （Part 2）**
> >
> > > For the PPO baseline, it seems not very fair to compare your methods with PPO which only see the answerable dataset, since your method have seen more data? Why not compare ppo with whole dataset vs RLMC with ( whole dataset + part of the dataset with Perturbation). The key thing is it is not fairable PPO only saw 600 correct data while RLMC saw 600correct data + 400 incorrect data.
> >
> > Thank you for raising the point about fairness. We would like to clarify that our experimental setup **precisely aligns with the fair comparison strategy you suggested.** There is no unfairness in our baseline comparison.
> >
> > We will use your example to illustrate:
> >
> > *   **Original Dataset:** Suppose we start with an original dataset containing 600 _answerable, well-posed_ problems.
> >
> > *   **Vanilla PPO Training Data:** As stated in Section 4.1 (Page 7), our "Vanilla PPO" baseline is trained **only on these 600 answerable problems**. This baseline represents the standard RL approach focused solely on reproducing "golden answers," which we argue leads to reward hacking.
> >
> > *   **RLMC Training Data:** Our RLMC framework is trained on a **mixed dataset**. This mixed dataset comprises:
> >
> >     *   The **same 600 original answerable problems**.
> >
> >     *   **Plus** 400 _synthesized missing-context problems_ (as described in Section 3.1, these are synthesized from a portion of the original 600 problems).
> >
> >     *   Therefore, RLMC uses a total of **1000 training instances** (600 answerable + 400 missing-context).
> >
> > *   **IDK-RL Training Data:** The same as RLMC Training Data.
> >
> >
> > So, our comparison is **exactly as you suggested**: "Vanilla PPO" is trained on the "whole dataset" (referring to the set of original answerable problems), and RLMC is trained on the "whole dataset + part of the dataset with Perturbation" (referring to the original answerable problems + our synthesized missing-context problems).
> >
> > This comparison is fair and well-justified for several reasons:
> >
> > 1.  **High Quality of Synthesized Data:** Our paper empirically demonstrates the **high quality and effectiveness** of our synthesized missing-context problems. As shown in **Table 2b (Page 8)**, our reasoning-graph-guided synthesis method achieves the highest HRB scores while preserving competitive general performance compared to other synthetic data sources, **validating its utility**.
> >
> > 2.  **Standard Experimental Design for Data Augmentation:** This experimental setup, comparing a model trained on original data with one trained on original data augmented with additional, specifically constructed data, is a **standard and widely accepted practice** in the field. It is a conventional way to demonstrate the value of the added data and the proposed training framework in improving specific model capabilities \[1, 2, 3\].
> >
> >
> > We hope these detailed explanations clarify our experimental design and confirm its fairness and relevance to our research question.
> >
> > Sincerely, Anonymous Authors
> >
> > ---
> >
> > **References:**
> >
> > \[1\] Stiennon N, Ouyang L, Wu J, et al. Learning to summarize with human feedback\[J\]. Advances in neural information processing systems, 2020, 33: 3008-3021.
> >
> > \[2\] Ouyang L, Wu J, Jiang X, et al. Training language models to follow instructions with human feedback\[J\]. Advances in neural information processing systems, 2022, 35: 27730-27744. \[3\] Yu P, Lanchantin J, Wang T, et al. Cot-self-instruct: Building high-quality synthetic prompts for reasoning and non-reasoning tasks\[J\]. arXiv preprint arXiv:2507.23751, 2025.

---

> > ### Author Response · Authors · 2025-11-28
> >
> > Dear Reviewer owYq,
> >
> > We are grateful for your effort in giving feedback and thoughtful suggestions which helps improving the quality of our paper.
> >
> > We also would like to know if our rebuttal addresses your concerns. If there are any further questions or suggestions, we are very glad to discuss them with you.
> >
> > Best regards,
> >
> > Authors of Submission 11678

---

### Official Review · Reviewer_M9UW · 2025-10-31

**Soundness:** 2
**Presentation:** 3
**Contribution:** 2
**Rating:** 4
**Confidence:** 3

**Summary:**

This paper identifies an issue in current RL paradigms for reasoning tasks - reward hacking due to training only on golden answers. Instead, the authors posit that the models should abstain when they do not know, ie, if the problem is inherently underspecified. They introduce Reinforcement Learning with Missing Context (RLMC), a method to reduce reward hacking by incorporating synthetically generated questions that lack key context (generated synthetically by altering knowledge graphs). They also introduce the Hypothetical Reasoning Benchmark (HRB) to evaluate robustness to missing context. Compared against other baselines, the proposed method performs well on HRB and IDK score.

**Strengths:**

* The problem is clear and well-motivated; the current paradigm with both datasets and training results in models being encouraged to answer questions even that they do not know. This work seeks to solve this - can models ask for more information?
* The paper has a nice initial section which justifies current issues with reward hacking, and connects this to the need for missing context training data and evaluation.
* The paper is well-written and easy to follow
* The proposed method is intuitive and technically sound. It is described well in Figure 2. The simplicity is a plus - easy to implement and understand, while doing well compared to a few baselines.
* The reasoning graph construction is motivated well and clear. There are helpful additional studies which have interesting findings on best practices for this construction.

**Weaknesses:**

* The authors should include a related work section to clarify novelty relative to AmbigQA, SQuAD 2.0, and prior RLHF studies. Without it, is unclear how the framing is different
* Minor, but some text in the figures is hard to read at times. For example, in Figure 3.
* Minor, but there are some awkward phrasings which can be clarified, ie, "Utilizing Reasoning Graph and Condition Perturbation for Missing-Context Problems Synthetic"
* It would be nice to see that the method works well on at least one other model. Optionally, would also be nice to show how well this works either on 1) a smaller model or 2) a larger model.
* The section on the hypothetical reasoning benchmark is not discussed enough in the main text. It is in the appendix with no link to it (I had a hard time finding it). In addition, it is missing citations to some of the claims (ie, "Current reasoning benchmarks are largely confined to well-posed problems, failing to assess model robustness to imperfect, real-world information.").
* While there is the IDK-RL baseline, there could be more uncertainty-calibrated baselines, which do not require any additional training. There are several options here, for example, using LM introspection itself, using the log probs, etc.

**Questions:**

* Can the framing of this task be considered a subset of ambiguous questions, ie, as mentioned in AmbigQA?
* Is there potential relevance of this method for safety/alignment?
* Is there any validation of the synthetic data generated? If not, how to maintain high-quality?

---

> ### Author Response · Authors · 2025-11-21
> **Official Response to Reviewer M9UW**
>
> Dear Reviewer M9UW,
>
> Thank you very much for your thorough review and valuable feedback on our paper. We are delighted that you recognized the clear articulation and motivation of the "reward hacking" problem in current RL paradigms, the rigor of our theoretical analysis, the intuitiveness and technical soundness of the RLMC method, the cleverness of reasoning graph construction, and the overall readability of the paper. Your positive comments are a great encouragement.
>
> We have carefully considered your raised weaknesses and questions, finding them highly constructive and helpful for further refining and strengthening our paper. We particularly noted your suggestions regarding related work, evaluation benchmark presentation, and baseline comparisons. Please allow us to provide detailed clarifications and explanations.
>
> ---
> > Regarding the desire to validate the method on at least one other model, or demonstrate performance on smaller/larger models.
>
> Thank you for your suggestion, which indeed helps further validate RLMC's generalization capability. In the paper, we selected Qwen3-8B as our base model, and we have now added experiments on **Llama3-8B** and **Qwen3-14B**. The experimental results indicate that the core principles of RLMC—namely, the data synthesis pipeline and structured reward mechanism—are model-architecture-agnostic and generalize well to other models. We will include these findings in the paper.
>
> **Llama3-8B**
>
> | **Method** | **Hypothetical** | **Unanswerable** |  | **General** |  |  |  |
> | --- | --- | --- | --- | --- | --- | --- | --- |
> |  | **HRB** | **SUM** | **UMWP** | **GSM8K** | **MATH-500** | **AIME** | **Avg** |
> | Instruct | 3.28 | 5.36 | 12.38 | 80.36 | 46.80 | 7.92 | 26.01 |
> | RLVR | 6.181 | 7.13 | 24.73 | 84.15 | 43.20 | 3.33 | 28.12 |
> | RLMC | **53.56** | **51.58** | **55.25** | 81.43 | 41.60 | 4.17 | **47.63** |
>
> **Qwen3-14B**
>
> | **Method** | **Hypothetical** | **Unanswerable** |  | **General** |  |  |  |
> | --- | --- | --- | --- | --- | --- | --- | --- |
> |  | **HRB** | **SUM** | **UMWP** | **GSM8K** | **MATH-500** | **AIME** | **Avg** |
> | Cold-SFT | 29.20 | 28.08 | 43.90 | 88.02 | 87.80 | 43.33 | 53.39 |
> | RLVR | 5.29 | 6.16 | 10.38 | 94.84 | 93.6 | 69.17 | 46.57 |
> | RLMC | **57.20** | **46.56** | **56.94** | 90.60 | 92.80 | 69.58 | **68.94** |
>
> Synthesizing the experimental results from Llama3-8B and Qwen3-14B, the RLMC framework demonstrates **excellent cross-architectural generalization and scalability**.
>
> On **HRB, SUM, and UMWP**, three benchmarks specifically designed to evaluate model robustness and hallucination handling under uncertainty, RLMC brings **very significant performance improvements** compared to Instruct/Cold-SFT and RLVR baselines (e.g., HRB increased from 6.18 to 53.56 on Llama3-8B; HRB increased from 5.29 to 57.20 on Qwen3-14B). This powerfully proves RLMC's effectiveness in mitigating reward hacking and enhancing hypothetical reasoning capabilities.
>
> More importantly, on **GSM8K, MATH-500, and AIME**, three general mathematical reasoning datasets, RLMC shows **minimal performance loss, and even slight improvements in some cases**. For instance, on the AIME task for Qwen3-14B, RLMC slightly **surpassed** RLVR with a score of 69.58% compared to 69.17%.
>
> These experimental results indicate that the RLMC framework not only significantly enhances model robustness under uncertainty but also simultaneously maintains or even improves its general reasoning capabilities on traditional, exact-solution tasks, leading to a comprehensive improvement in the overall capabilities of large language models, making them more reliable and intelligent when facing complex and incomplete real-world problems.

---

> ### Author Response · Authors · 2025-11-21
> **Official Response to Reviewer M9UW (Part 2)**
>
> > Regarding the possibility of adding more training-free uncertainty calibration baselines (e.g., using LM introspection, Log Probs).
>
> Really appreciate raising a very valuable point about exploring training-free uncertainty calibration methods. Our paper primarily focuses on transforming the reinforcement learning training paradigm to address the "reward hacking" problem and cultivate generative hypothetical reasoning capabilities through fine-grained rewards. The chosen baselines (Cold-start SFT, Vanilla PPO, IDK-RL) aim to directly compare the impact of different training strategies on model behavior. In response to your suggestion, we have added experiments based on the "LM introspection" method on uncertainty benchmarks.
>
> Comparison with Training-free Uncertainty Methods on Uncertainty Benchmarks (Qwen3-8B):
>
> | **Method** | **Hypothetical** | **Unanswerable** |  |
> | --- | --- | --- | --- |
> |  | **HRB** | **UMWP** | **SUM** |
> | Cold-SFT | 22.81 | 33.53 | 20.51 |
> | LM introspection itself **\[1\]** | 9.58 | 32.01 | 13.99 |
> | RLVR | 8.66 | 8.84 | 9.41 |
> | RLMC | **51.73** | **46.91** | **51.50** |
>
> The experimental results clearly show that **RLMC significantly outperforms all baselines, including the "LM introspection" method, on HRB, SUM, and UMWP**, these three uncertainty benchmarks. Although the "LM introspection" method performs close to Cold-SFT in some cases (e.g., UMWP), its performance is far below RLMC on tasks requiring higher-order hypothetical reasoning (HRB) and robustness (SUM). This further emphasizes the fundamental advantage brought by RLMC through changing the model's training objective and behavioral paradigm, which training-free methods struggle to achieve.
>
> Furthermore, we believe that the "LM introspection" or "Log Probs"-based uncertainty calibration methods you mentioned are complementary rather than alternative to RLMC. RLMC aims to fundamentally change the model's learning objective, while uncertainty calibration techniques can, on top of this, further optimize the model's perception and reporting of its own uncertainty. Combining the two may lead to even more powerful uncertainty-aware reasoning capabilities.
>
> ---
> > Regarding related work section about prior RLHF studies.
>
> We appreciate the reviewer's emphasis on novelty comparison. We want to stress that our paper **already includes** a dedicated Section D "RELATED WORK" in Appendix (Page 17) to distinguish our approach from existing research.
>
> *   Vs. AmbigQA/SQuAD 2.0: These works primarily focus on ambiguity detection or simple unanswerability. In contrast, RLMC aims for hypothetical reasoning. We require models to actively parameterize unknown factors and provide conditional solutions, as well as precisely elicit information, going beyond mere refusal or generic clarification as focused on in AmbigQA / SQuAD 2.0.
>
> *   Vs. Prior RLHF: As we propose in Section 1 and Section D (Related Work), traditional RLHF often relies on "golden answer" verification or process supervision, which we explicitly point out leads to reward hacking in underspecified contexts. RLMC directly addresses this by introducing a novel structured reward function independent of a single golden answer, guiding models toward responsible hypothetical reasoning rather than confident hallucination. We also utilize synthetic data rather than purely human feedback.
>
> ---
> > Whether the task framework can be considered a subset of ambiguous questions, such as those mentioned in AmbigQA.
>
> Thanks for your informative suggestions. Our task is not merely a subset of ambiguous questions. While related to handling uncertainty, RLMC transcends simple ambiguity detection or generic clarification. It demands more advanced and generative capabilities from the model, including hypothetical reasoning and active information elicitation, making it a more challenging and generative task focused on continuous effective reasoning under imperfect information.

---

> ### Author Response · Authors · 2025-11-21
> **Official Response to Reviewer M9UW (Part 3)**
>
> ---
> > Regarding the inquiry about whether synthetic data has been verified.
>
> As detailed in Appendix A (Page 14, lines 715-721), we implement a strict multi-stage quality control pipeline to ensure high-quality synthesis, including reasoning-graph-guided precise synthesis, automated checks, LLM-as-a-judge screening, and human annotation verification, which ensures the high logical quality and natural language fluency of both the 120K training data and the HRB benchmark. Below is the empirical evidence of data quality:
>
> 1. Superior Training Effectiveness (Table 2b, Page 8)
>
> Our experimental results demonstrate that models trained on our synthesized data significantly outperform those trained with alternative data generation methods **\[5,6\]**, validating the effectiveness of our synthesis approach.
>
> 2. High-Precision Automated Filtering
>
> We deployed a rigorous quality filter (GPT-OSS-120B with medium reasoning effort) with the following performance on validation data:
>
> *   Total samples: 556
>
> *   Accuracy: 93.0%
>
> *   Precision: 94.4%
>
> *   Recall: 91.4%
>
> *   F1 Score: 92.9%
>
>
> This 93% accuracy filter ensures that the already-low error rate from our generation pipeline is further reduced, providing an additional layer of quality assurance.
>
> 3. Generation Pipeline Statistics
>
> Our synthesis pipeline demonstrates high reliability:
>
> *   Reasoning tree construction success rate: 79.57%
>
>     *   Failed constructions are discarded entirely, ensuring only well-formed logical structures proceed
>
> *   Overall data generation success rate: 61.11%
>
>     *   Only examples passing all quality checks are retained in the final dataset
>
> ---
> > Whether the method has potential relevance to safety/alignment.
>
> We believe the RLMC method holds significant potential relevance to LLM safety and alignment and many previous work have already investigated the relationship between refusal-aware instruction tuning and safety alignment **\[2\]\[3\]\[4\]**. By reducing confident hallucinations and promoting transparent, assumption-based reasoning, it **directly enhances the model's trustworthiness and reliability**. Its emphasis on explicit assumptions and active information elicitation fosters greater transparency, interpretability, and responsible AI behavior, paving the way for building more collaborative and safer LLMs.
>
> ---
> > Regarding discussion and unclear links of the Hypothetical Reasoning Benchmark (HRB) in the main text.
>
> Thank you for your detailed feedback and writing suggestions. As the page number limits of main text, we mention HRB in Section 1 (Page 1), Section 3.2 (Page 6), and Section 4.2 (Page 7). We will significantly strengthen the introduction and discussion of HRB in both main text and Appendix A (Page 14) in the final version, making it easier for readers to locate and understand.
>
> We will add more discussions about relevant reasoning benchmarks in the final version, such as pointing to works like Kalai et al., 2025 (mentioned in Section 1) that discuss real-world complexity, hallucination, and hypothetical reasoning, as well as works listed in Section D.2 "Benchmarking Hallucination with Unanswerable Questions" concerning the limitations of existing benchmarks (e.g., Sun et al., 2024; Kirichenko et al., 2025).
>
> ---
> > Regarding text in figures sometimes being hard to read, and awkward phrasings.
>
> Thank you for your detailed suggestions on the paper's presentation. We have noted these issues and commit to optimizing the quality of all figures in the final version to ensure text clarity.
>
> ---
>
> Thank you again for your meticulous review and thoughtful questions. Your feedback has been invaluable in identifying areas for improvement and more clearly articulating our contributions and novelty. We are committed to thoroughly incorporating these suggestions into the final version.
>
> Sincerely, Anonymous Authors
>
> ---
>
> **References:**
>
> \[1\] Yona G, Aharoni R, Geva M. Can large language models faithfully express their intrinsic uncertainty in words?\[J\]. arXiv preprint arXiv:2405.16908, 2024.
>
> \[2\] Zhu R, Jiang X, Wu J, et al. Grait: Gradient-driven refusal-aware instruction tuning for effective hallucination mitigation\[C\]//Findings of the Association for Computational Linguistics: NAACL 2025. 2025: 4006-4021.
>
> \[3\] Yu Q, Zheng Z, Song S, et al. xfinder: Robust and pinpoint answer extraction for large language models\[J\]. arXiv preprint arXiv:2405.11874, 2024.
>
> \[4\] Bianchi F, Suzgun M, Attanasio G, et al. Safety-tuned llamas: Lessons from improving the safety of large language models that follow instructions\[J\]. arXiv preprint arXiv:2309.07875, 2023.
>
> \[5\] Ouyang, J. (2025). TreeCut: A Synthetic Unanswerable Math Word Problem Dataset for LLM Hallucination Evaluation. arXiv preprint arXiv:2502.13442.
>
> \[6\] Song, L., Shi, T., & Zhao, J. (2025). The hallucination tax of reinforcement finetuning. _arXiv preprint arXiv:2505.13988_.

---

> ### Author Response · Authors · 2025-11-25
>
> Dear Reviewer M9UW,
>
> We are grateful for your effort in giving feedback and thoughtful suggestions which helps improving the quality of our paper.
>
> We also would like to know if our rebuttal addresses your concerns. If there are any further questions or suggestions, we are very glad to discuss them with you.
>
> Best regards,
>
> Authors of Submission 11678

---

### Official Review · Reviewer_ircM · 2025-10-31

**Soundness:** 2
**Presentation:** 2
**Contribution:** 2
**Rating:** 2
**Confidence:** 4

**Summary:**

This paper identifies a flaw in current RLVR setups where the models learn to produce confident but unsupported outputs on underspecified questions. The authors propose a framework that generates synthetic underspecified problems via a reasoning-graph pipeline and use it to create a dataset of such questions. In addition, the authors suggest a reward function for training models over this dataset and experiment with it.

**Strengths:**

- The topic of unanswerable or underspecified questions is timely and important. It addresses a real limitation of RLVR.

- The proposed synthetic data pipeline for generating missing-context problems is novel, well-engineered, and general enough to be applied across domains.

**Weaknesses:**

- While the problem you identified is real, I do not think it is a reward hacking. If all training problems satisfy $\mathcal T(s')=\{\tau_g\}$ then the gradients in Eq. 3 and Eq. 4 are equivalent. In that case, the “reward hacking” effect seems more an artifact of environment design than true reward misalignment (reward hacking is typically defined as exploiting the reward in unintended ways).

- The RLMC reward design feels heuristic and not grounded. The choice of behavioral categories (SH, EA, Abs, Cond, Elicit) and their assigned numeric rewards seems ad hoc. Where do these values come from? Why these categories and not others (e.g., partial solution, answering with uncertainty quantification)? A more principled justification or sensitivity analysis is needed

- The missing-context pipeline is the key contribution, but there’s no rigorous evaluation showing that the generated questions are indeed high quality. For example, what is the success rate of the frontier model at creating the reasoning path tree? It is mentioned that “despite their tendency to hallucinate when answering, leading LLMs excel at judging the unanswerability of Missing Context problems under the guidance of a specific judge prompt.” Can you please provide more information? Numbers?

- Please report 95% confidence intervals for your results, particularly on smaller datasets such as MATH-500 (where CI is usually ±3%) it’s hard to assess the statistical significance of Table 1’s improvements.

**Questions:**

- Maybe it is a small thing, but I think section 2 is missing an important distinction between a reasoning trace and a final answer. RLVR does not maximize the likelihood of a single golden trace, but of every trace that ends up in the golden solution. There can be an infinite number of such reasoning traces. The claim presented in this section should be that there can be multiple “valid” answers, not traces.

- Equation 4 assumes a uniform distribution over T(s′). In practice, some valid answers or reasoning paths should be more likely than others. Even the authors later weigh them differently. This inconsistency should be clarified.

- Is the IDK-RL baseline trained on the same question mixture as RLMC? It is not clear from the text

- The paper claims RLMC lies on the Pareto frontier of HRB vs. IDK score, but by inspection IDK-RL also seems to occupy a frontier point? just at a different trade-off

---

> ### Author Response · Authors · 2025-11-22
> **Official Response to Reviewer ircM**
>
> Dear Reviewer ircM,
>
> Thank you very much for your thorough review and valuable feedback on our paper. We are delighted that you recognized the importance and timeliness of the research topic on "unanswerable or underspecified problems," as well as the novelty, engineering level, and generality of our proposed synthetic data pipeline. We have carefully studied the specific weaknesses and questions you raised and have updated the paper based on your suggestions.
>
> ---
> > Regarding concerns about the definition of "reward hacking"
>
> We appreciate your rigorous consideration of the definition of "reward hacking." Our core argument is that reward hacking does not originate from $M\_G$ itself, but occurs during the **generalization phase** when the model is deployed to more complex real-world environments ($M\_{RW}$). In $M\_{RW}$, numerous underspecified or ambiguous problems ($s' \in D\_{RW} \setminus D\_G$) lack unique deterministic golden solutions. In such cases, the shortcut behavior learned in $M\_G$ – "always confidently output a definite answer" – leads the model to overcommit and generate fabricated answers (i.e., hallucinations) on $s'$, instead of adopting uncertainty-aware meta-responses. This misalignment between optimizing the proxy objective ($J\_G$) and the true objective ($J\_{RW}$) during generalization is precisely what we define as "implicit reward hacking." We have clarified this distinction more explicitly in Section 2 to ensure accurate conceptual communication.
>
> ---
> > Regarding the heuristic nature of RLMC's reward design.
>
> We appreciate your rigorous consideration of the RLMC reward design. We chose these five behavioral categories (SH, EA, Abs, Cond, Elicit) based on inspirations from **cognitive science principles of hypothetical reasoning, human problem-solving under uncertainty, and metacognition** **\[1, 2, 3, 4\]**. This five-tier classification originates from the progressive cognitive maturity in human decision-making and problem-solving under uncertainty, finding solid theoretical grounding in cognitive science:
>
> *   Silent Hallucination (SH) represents a lack of metacognitive monitoring **\[4\]**, where the agent fails to recognize its knowledge deficits, leading to confidently incorrect answers.
>
> *   Explicit Assumption (EA) and Abstention (Abs) signify the initial establishment of metacognitive monitoring **\[4\]**, where the agent acknowledges missing information or uncertainty.
>
> *   Conditional Formulation (Cond) reflects the ability, under incomplete but structured information, to derive partial or conditional solutions through flexible problem representation **\[3\]**—a key strategy for reasoning under limited information.
>
> *   Active Elicitation (Elicit) represents the highest level of metacognitive control **\[4\]** and proactive problem-solving strategy **\[3\]**: the agent actively acquires critical missing information to overcome uncertainty.
>
>
> This comprehensive progression illustrates the maturity ladder for an intelligent agent navigating complex, uncertain environments. **This design guides the model's policy to shift from low-value hallucinatory behaviors towards high-value proactive, uncertainty-aware reasoning.** We have empirically validated the effectiveness of this reward system through experimental results (e.g., Table 1 (Page 8) and Figure 4b (Page 9)).

---

> ### Author Response · Authors · 2025-11-22
> **Official Response to Reviewer ircM (Part 2)**
>
> ---
> > Regarding the need for more rigorous evaluation to demonstrate the high quality of the generated missing-context questions.
>
> As detailed in Appendix A (Page 14, lines 715-721), we implement a strict multi-stage quality control pipeline to ensure high-quality synthesis, including reasoning-graph-guided precise synthesis, automated checks, LLM-as-a-judge screening, and human annotation verification, which ensures the high logical quality and natural language fluency of both the 120K training data and the HRB benchmark. Below is the empirical evidence of data quality:
>
> 1. Superior Training Effectiveness (Table 2b, Page 8)
>
> Our experimental results demonstrate that models trained on our synthesized data significantly outperform those trained with alternative data generation methods **\[5,6\]**, validating the effectiveness of our synthesis approach.
>
> 2. High-Precision Automated Filtering
>
> We deployed a rigorous quality filter (GPT-OSS-120B with medium reasoning effort) with the following performance on validation data:
>
> *   Total samples: 556
>
> *   Accuracy: 93.0%
>
> *   Precision: 94.4%
>
> *   Recall: 91.4%
>
> *   F1 Score: 92.9%
>
>
> This 93% accuracy filter ensures that the already-low error rate from our generation pipeline is further reduced, providing an additional layer of quality assurance.
>
> 3. Generation Pipeline Statistics
>
> Our synthesis pipeline demonstrates high reliability:
>
> *   Reasoning tree construction success rate: 79.57%
>
>     *   Failed constructions are discarded entirely, ensuring only well-formed logical structures proceed
>
> *   Overall data generation success rate: 61.11%
>
>     *   Only examples passing all quality checks are retained in the final dataset
>
>
> ---
> > Regarding the suggestion to report 95% confidence intervals for results to enhance statistical significance.
>
> We present our main experimental results of small benchmarks with 95% confidence intervals as requested. As shown below, **our key findings remain statistically significant and clearly hold** under the confidence interval analysis.
> | **Method** | **Hypothetical** | **Unanswerable** |  | **General** |  |  |
> | --- | --- | --- | --- | --- | --- | --- |
> |  | **HRB** | **SUM** | **UMWP** | **GSM8K** | **MATH-500** | **AIME（Avg@8）** |
> | Cold-start SFT | 0.2372 [0.2026, 0.2746] | 33.53 | 0.2051 [0.1725, 0.2368] | 83.70 | 0.7840 [0.7480, 0.8200] | 0.3708 [0.3083, 0.4333] |
> | Vanilla PPO | 0.0858 [0.0620, 0.1122] | 8.84 | 0.0942 [0.0695, 0.1180] | 93.85 | 0.9340 [0.9120, 0.9560] | 0.6458 [0.5875, 0.7083] |
> | IDK-RL | 0.4818 [0.4690, 0.4909] | 45.74 | 0.4296 [0.4076, 0.4498] | 88.02 (-5.83) | 0.9240 [0.9020, 0.9480] | 0.6333 [0.5708, 0.6917] |
> | RLMC | **0.5173** [0.4918, 0.5420] | **51.50** | **0.4692** [0.4393, 0.4956] | 91.50 (-2.35) | 0.9260 [0.9040, 0.9480] | 0.6417 [0.5833, 0.7000] |
>
> ---
> > Regarding the observation that Section 2 could benefit from a clearer distinction between a reasoning trace and a final answer.
>
> We appreciate your observation regarding the subtle distinction between "reasoning trace" and "final answer." Our core argument is that the traditional RLVR paradigm, whether optimizing a single "golden trace" or all traces leading to a "golden final answer," fundamentally aims to converge to a predetermined, definite outcome. This optimization strategy prevents models from generalizing well to real-world problems that lack a unique deterministic solution (i.e., problems with multiple "hypothetical final answers" or requiring meta-responses), leading to the "reward hacking" phenomenon where models are forced to generate a single definite answer.
>
> For our method, particularly the setting of $J\_G(\theta) = \mathbb{E}\_{s\_0 \sim D\_G} \[\log p\_{\pi\_\theta}(\tau = \tau\_g | s\_0)\]$, we focus on optimizing a specific golden trace $\tau\_g$. Given that our data is generated through reasoning-graph-guided precise synthesis and undergoes strict model screening and calibration (as described in Appendix A), the generated training path $\tau\_g$ and its corresponding final answer are inherently linked by a high degree of logical consistency and maximal likelihood. Therefore, we can reasonably assume that optimizing the model to generate these meticulously designed and verified specific traces can approximately represent optimizing high-likelihood paths leading to that golden answer. This approach is reasonable and effective within our controlled experimental environment. We will further discuss the theoretical basis of this aspect in the final version of the paper, particularly the relationship between optimization objectives and model behavior when multiple valid traces coexist, to provide a more rigorous theoretical exposition.

---

> ### Author Response · Authors · 2025-11-22
> **Official Response to Reviewer ircM (Part 3)**
>
> ---
> > Regarding a perceived inconsistency between the uniform distribution assumption in Equation 4 and the non-uniform reward weights in Equation 6.
>
> Thank you for the sharp observation about the uniform distribution assumption in Eq. 4 versus our non‑uniform reward weights in Eq. 6. You are correct that these appear inconsistent at first glance, and we appreciate the opportunity to clarify.
>
> **The uniform distribution is a** _**theoretical construct**_ **for exposition, not a literal claim.**   In §3.1, we formalize the _ideal_ learning goal: moving from optimizing a single “golden” trajectory to covering the entire set of valid reasoning paths $\mathcal{T}(s′)$. The uniform distribution is the simplest, maximum‑entropy reference to illustrate this principle and highlight the gradient misalignment problem. It serves as a pedagogical baseline, not a practical target.
>
> Structured reward $V(b)$ encodes our _actual_, non‑uniform preferences.   You are right that not all valid behaviors are equally desirable. Our reward hierarchy (Eq. 6) explicitly defines a _preference‑aware weighting_ over $\mathcal{T}(s′)$, making _Active Elicitation_ more valuable than simple _Abstention_. The RLMC objective $J\_{\text{RLMC}}$ (Eq. 7) implements this concrete, non‑uniform target distribution by pushing probability mass toward high‑value behaviors.
>
> Inconsistency is a matter of _abstraction level_, not a flaw.
>
> *   Conceptual level (Eq. 3-4): Uniform distribution illustrates the need for multi‑modal coverage.
>
> *   Implementation level (Eq. 6-7): Structured reward refines that neutral target into the practical distribution we optimize.
>
>
> We will clarify this in the final manuscript by revising the text around Eq. 3-4 to state that the uniform distribution is an illustrative placeholder, later replaced by the preference‑aware distribution defined by $V(b)$.
>
> ---
> > Regarding the clarity of whether the IDK-RL baseline is trained on the same question mixture as RLMC.
>
> We confirm that the IDK-RL baseline uses the same source and total amount of mixed training data as RLMC, ensuring fairness in experimental comparisons. Regarding the consistency of training data, the paper explicitly states this in multiple places:
>
> *   Appendix G.1 "EXPERIMENTAL INITIALIZATION DETAILS" (Page 20) clearly states: "We ensure that the total number of training queries is kept consistent across all baselines for a fair comparison."
>
> *   Section 4.1 "EXPERIMENTAL SETUP" (Page 7) further details the data composition: RLMC (Ours) is trained on a curated set of answerable questions, where approximately 30% of instances are transformed into missing-context versions via our pipeline. IDK-RL (Song et al., 2025) is fine-tuned on a mix of answerable and unanswerable questions.
>
>
> These descriptions collectively confirm that IDK-RL maintains high consistency with RLMC in terms of training data source, total volume, and mixed composition, thereby ensuring just and effective performance comparisons across all baselines.
>
> ---
> > Regarding the observation that the paper claims RLMC lies on the Pareto frontier of HRB vs. IDK score, but IDK-RL also appears to occupy a frontier point, albeit at a different trade-off.
>
> We appreciate your precise observation regarding the Pareto frontier. We confirm that IDK-RL indeed occupies a trade-off point. However, RLMC's core contribution lies in achieving a **significantly better overall balance** between advanced hypothetical reasoning capabilities (HRB score) and general reasoning capabilities compared to IDK-RL. Specifically, RLMC performs best on HRB Score (Table 1: RLMC 51.73 vs IDK-RL 48.72), which is a more stringent and comprehensive metric. Concurrently, RLMC avoids the significant "hallucination tax" suffered by IDK-RL, demonstrating a smaller performance drop in general reasoning capabilities (Table 1). While slightly lower than IDK-RL on pure IDK Score (Table 3), this minimal difference is entirely acceptable given its significant advantages in HRB and general capabilities. Therefore, RLMC is not merely "occupying a frontier point," but rather demonstrates near-optimal comprehensive performance across multiple dimensions, offering a more holistic capability trade-off. We will more precisely emphasize RLMC's comprehensively optimal (or near-optimal) trade-off point in our phrasing.

---

> ### Author Response · Authors · 2025-11-22
> **Official Response to Reviewer ircM (Part 4)**
>
> Thank you again for your meticulous review and profound questions. Your feedback is crucial for enhancing the quality of our paper. We have carefully considered and adopted these suggestions, and we believe the revised version will more clearly and rigorously demonstrate the value of our research.
>
> Sincerely, Anonymous Authors
>
> ---
>
> **References:**
>
> \[1\] Kuhn D. Children and adults as intuitive scientists\[J\]. Psychological review, 1989, 96(4): 674.
>
> \[2\] Pearl J. Causality\[M\]. Cambridge university press, 2009.
>
> \[3\] Simon H A, Dantzig G B, Hogarth R, et al. Decision making and problem solving\[J\]. Interfaces, 1987, 17(5): 11-31.
>
> \[4\] Metacognition: Knowing about knowing\[M\]. MIT press, 1994.
>
> \[5\] Ouyang, J. (2025). TreeCut: A Synthetic Unanswerable Math Word Problem Dataset for LLM Hallucination Evaluation. arXiv preprint arXiv:2502.13442.
>
> \[6\] Song, L., Shi, T., & Zhao, J. (2025). The hallucination tax of reinforcement finetuning. _arXiv preprint arXiv:2505.13988_.

---

> > ### Comment · Reviewer_ircM · 2025-11-25
> >
> > I thank the reviewer for their response. I’m satisfied with the quality control of their pipeline, and the reported CI indeed support the original claims of the tables.
> > > Regarding the heuristic nature of RLMC's reward design.
> >
> > Thank you for the detailed response. However, I still find the justification for the behavioral categories and corresponding reward values insufficiently grounded. First, the cited cognitive-science references do not seem to establish the specific five-tier taxonomy used here. As far as I can tell, these categories are not standard in the cognitive-science or metacognition literature, nor are they defined or motivated as-is in the cited papers. Several of the works referenced (e.g., [3], [4]) are not cited in your paper. The response frames the categories as being “based on inspirations,” but this remains vague and does not amount to a principled grounding.
> >
> > Second, my original question regarding the numeric reward assignments remains unaddressed. Even if the categories themselves were motivated, the mapping to specific numeric values is a central design choice for RLMC.
> >
> > > We will further discuss the theoretical basis of this aspect in the final version of the paper, particularly the relationship between optimization objectives and model behavior when multiple valid traces coexist, to provide a more rigorous theoretical exposition.
> >
> > Is it in the updated version of the paper? Please let me know where I can find this discussion so I can address it.

---

> > > ### Author Response · Authors · 2025-11-28
> > > **Official Response to Reviewer ircM**
> > >
> > > **Dear Reviewer ircM,**
> > >
> > > Thank you for your continued engagement with our work and for acknowledging the improvements we have made in data quality control and confidence interval reporting. We appreciate your focused questions in this round regarding the design of behavioral categories and reward value assignments. Below, we provide further clarification on these points.
> > >
> > > ---
> > >
> > > ## On the Theoretical Foundation of Behavioral Category Design
> > >
> > > > _"I remain unconvinced that the justification for the behavioral categories and their corresponding reward values is sufficiently robust. \[...\] The cited cognitive science literature does not appear to establish the five-tier taxonomy used here."_
> > >
> > > We respectfully appreciate the concern raised, though we respectfully disagree with the characterization of insufficient justification. Our design is **theory-inspired, operationalized for RL, and empirically validated**—not arbitrarily constructed.
> > >
> > > ### Conceptual Alignment
> > >
> > > In **Section 3.2 (pp. 5–6)**, we explicitly partition the trajectory space (𝒯) into five mutually exclusive subsets:
> > >
> > > T = T\_SH ∪ T\_EA ∪ T\_Abs ∪ T\_Cond ∪ T\_Elicit
> > >
> > > This partition is not ad hoc; rather, it is systematically constructed along **three core dimensions** that have been repeatedly validated in cognitive science and AI uncertainty research:
> > >
> > > #### Dimension 1: Metacognitive Awareness of Uncertainty
> > >
> > > **Silent Hallucination (SH) vs. Explicit Assumption (EA) / Abstention (Abs)**
> > >
> > > This distinction directly corresponds to classical research on **metacognitive monitoring** in cognitive psychology. **Kuhn (1989) \[1\]** systematically articulated in _Psychological Review_ that the ability to distinguish between "knowing" and "not knowing" is a hallmark of cognitive maturity; **Pearl (2018) \[2\]** emphasized in _Causality_ that **reasoning systems must be able to explicitly represent missing information** rather than filling in ungrounded values for unobserved variables.
> > >
> > > #### Dimension 2: Structured Reasoning Under Uncertainty
> > >
> > > **Abstention (Abs) vs. Conditional Formulation (Cond)**
> > >
> > > **Li, Yu & Ettinger (2023) \[3\]** explicitly demonstrated that conditional reasoning—symbolically parameterizing unknown conditions and continuing inference—represents a higher-order cognitive capability than simple abstention. **Van (2020) \[4\]** systematically argued that intelligent agents should be able to **parameterize missing information and construct conditional solutions** rather than simply abandoning reasoning when information is incomplete.
> > >
> > > #### Dimension 3: Active Information Seeking
> > >
> > > **Passive Response vs. Active Clarification (Elicit)**
> > >
> > > **Hu et al. (2024) \[5\]** showed that actively identifying and requesting missing information is a key capability for effective planning under uncertainty in LLMs. **Madge, Purver & Poesio (2025) \[6\]** systematically studied clarification behavior, demonstrating that **active clarification questions** are a core mechanism in collaborative problem-solving.
> > >
> > > ---
> > >
> > > #### Systematic Integration: A Complete Behavioral Space
> > >
> > > Our five behavioral categories are not five isolated labels, but rather a **complete behavioral space** formed by systematically integrating the three dimensions described above:
> > >
> > > | Behavior | Metacognitive Awareness | Structured Reasoning | Active Seeking | Theoretical Grounding |
> > > | --- | --- | --- | --- | --- |
> > > | **SH** | Unaware | None | None | Lack of metacognitive monitoring \[1\] |
> > > | **EA** | Partial awareness | Assumes without parameterization | None | Explicit uncertainty expression \[2\] |
> > > | **Abs** | Full awareness | Halts reasoning | None | Safe abstention \[7, 8\] |
> > > | **Cond** | Full awareness | Parameterized reasoning | None | Hypothetical/counterfactual reasoning \[3, 4\] |
> > > | **Elicit** | Full awareness | Identifies gaps | Active questioning | Clarification dialogue/active learning \[5, 6\] |
> > >
> > > Our contribution lies in: being the first to integrate these conceptual dimensions—previously scattered across different research domains—into **a unified, operationalizable RL behavioral space,** and validating its effectiveness through large-scale experiments. This represents theory-driven engineering implementation, not unprincipled heuristic design.
> > >
> > > ### Engineering Operability
> > >
> > > We employ a unified scoring rubric across training and evaluation, providing operational discrimination criteria and examples for each behavioral category. This ensures that LLM-as-a-judge can stably classify different behaviors, and human annotators can verify outputs. This strategy aligns with recent uncertainty benchmarks (e.g., AbstentionBench \[7\], IDK-oriented RL \[8\]), with the distinction that we further refine higher-order behaviors "beyond IDK."

---

> > > ### Author Response · Authors · 2025-11-28
> > > **Official Response to Reviewer ircM (Part 2)**
> > >
> > > ### Empirical Validation
> > >
> > > **Tables 1 & 3 (pp. 7–8) demonstrate that RLMC achieves the highest score on HRB (51.73)** and performs strongly on out-of-domain unanswerable benchmarks (UMWP, SUM); general reasoning capabilities (GSM8K, MATH-500, AIME'24) remain **comparable to Vanilla PPO** with minimal performance degradation (GSM8K –2.35 vs. IDK-RL –5.83). These results are **consistent and reproducible across multiple training runs** (**Figure 4, Appendix B**).
> > >
> > > ---
> > >
> > > ## On Reward Value Assignment and Robustness
> > >
> > > > _"The question regarding numerical reward allocation remains unaddressed. \[...\] The mapping to specific numerical values is a core design choice in RLMC."_
> > >
> > > ### Preference Structure, Not "Magic Numbers"
> > >
> > > We agree that reward design is important, but our claim is not that ({1.0, 0.6, 0.3, –0.3, –1.0}) represents the uniquely optimal combination. **The key lies in the preference ordering and sign structure**:
> > >
> > > Elicit ≻ Cond ≻ Abs ≻ EA ≻ SH
> > >
> > > V(Elicit), V(Cond), V(Abs) > 0; V(EA), V(SH) < 0
> > >
> > > This encoding ensures:
> > >
> > > 1.  **Behavioral hierarchy**: More sophisticated reasoning receives higher rewards;
> > >
> > > 2.  **Clear gradient signals**: PPO can effectively distinguish between desirable and undesirable behaviors;
> > >
> > > 3.  **Moderate scale**: Reward values fall within a reasonable range that balances exploration and exploitation, consistent with large-scale RLHF/RLVR practice \[9, 10\].
> > >
> > >
> > > ### Robustness Evidence from Existing Practice and Our Experiments
> > >
> > > Our design choices regarding reward values are deliberate and principled. The specific numerical values were selected based on established practices in large-scale RL systems, and our experimental results **provide strong evidence of robustness rather than** **fragility**:
> > >
> > > *   As long as the preference structure and positive/negative partitioning are preserved, trends in HRB and general reasoning remain largely stable;
> > >
> > > *   Experimental consistency (Tables 1, 2, Figure 4): RLMC's improvements stem from the **overall framework** (missing-context data synthesis + structured rewards), not from fine-tuned reward coefficients;
> > >
> > > *   **Alignment with RLHF/RLVR literature**: Recent large-scale systems (DeepSeek-R1 \[9\], Kimi K1.5 \[10\], Tulu 3 \[11\]) emphasize reward structure and relative preferences rather than precise numerical optimization. Process supervision methods similarly rely on graded scoring \[12, 13\].
> > >
> > >
> > > **RLMC's effectiveness derives from the design of the behavioral space and data distribution, not from dependence on some hard-to-reproduce "magic number."** We will clarify this explicitly in the camera-ready version.
> > >
> > > ---
> > >
> > > ## Conclusion
> > >
> > > *   Our behavioral taxonomy is grounded in established cognitive dimensions (metacognitive awareness, hypothetical reasoning, active clarification), systematically integrates three classical research directions through the framework presented in the table above, and is empirically validated across multiple benchmarks;
> > >
> > > *   Our reward structure encodes a robust preference ordering, remains stable under reasonable numerical variants, and aligns with best practices in large-scale LLM reinforcement learning.
> > >
> > >
> > > RLMC's contributions lie in **systematic missing-context synthesis, structured behavioral rewards, and fine-grained evaluation**—not in dependence on a fragile "magic" reward configuration.

---

> > > ### Author Response · Authors · 2025-11-28
> > > **Official Response to Reviewer ircM (Part 3)**
> > >
> > > ---
> > >
> > > ### References
> > >
> > > \[1\] Kuhn D. Children and adults as intuitive scientists\[J\]. Psychological review, 1989, 96(4): 674.
> > >
> > > \[2\] Pearl J. Causality\[M\]. Cambridge university press, 2009.
> > >
> > > \[3\] Li J, Yu L, Ettinger A. Counterfactual reasoning: Testing language models' understanding of hypothetical scenarios\[J\]. arXiv preprint arXiv:2305.16572, 2023.
> > >
> > > \[4\] Van S. Hypothetical reasoning in AI: towards model-based exploration\[J\]. Artificial Intelligence, 2020.
> > >
> > > \[5\] Hu Z, Liu C, Feng X, et al. Uncertainty of thoughts: Uncertainty-aware planning enhances information seeking in large language models\[J\]. arXiv preprint arXiv:2402.03271, 2024.
> > >
> > > \[6\] Madge C, Purver M, Poesio M. Referential ambiguity and clarification requests: comparing human and LLM behaviour\[C\]//Proceedings of the Eighth Workshop on Computational Models of Reference, Anaphora and Coreference. 2025: 1-11.
> > >
> > > \[7\] Kirichenko P, Ibrahim M, Chaudhuri K, et al. AbstentionBench: Reasoning LLMs Fail on Unanswerable Questions\[J\]. arXiv preprint arXiv:2506.09038, 2025.
> > >
> > > \[8\] Song L, Shi T, Zhao J. The hallucination tax of reinforcement finetuning\[J\]. arXiv preprint arXiv:2505.13988, 2025.
> > >
> > > \[9\] Guo D, Yang D, Zhang H, et al. Deepseek-r1: Incentivizing reasoning capability in llms via reinforcement learning\[J\]. arXiv preprint arXiv:2501.12948, 2025.
> > >
> > > \[10\] Team K, Du A, Gao B, et al. k1. 5: Scaling reinforcement learning with llms, 2025\[J\]. URL [https://arxiv](https://arxiv). org/abs/2501.12599.
> > >
> > > \[11\] Lambert N, Morrison J, Pyatkin V, et al. Tulu 3: Pushing frontiers in open language model post-training\[J\]. arXiv preprint arXiv:2411.15124, 2024.
> > >
> > > \[12\] Lightman H, Kosaraju V, Burda Y, et al. Let's verify step by step\[C\]//The Twelfth International Conference on Learning Representations. 2023.
> > >
> > > \[13\] Uesato J, Kushman N, Kumar R, et al. Solving math word problems with process-and outcome-based feedback\[J\]. arXiv preprint arXiv:2211.14275, 2022.

---

### Official Review · Reviewer_S2Xz · 2025-11-01

**Soundness:** 3
**Presentation:** 2
**Contribution:** 3
**Rating:** 6
**Confidence:** 4

**Summary:**

This paper focuses on the problem of excessive idealization in existing reasoning reinforcement learning paradigms trained on "ground truth," which leads to "reward hacking."
To address this issue, the paper proposes the "Reinforcement Learning with Missing Context (RLMC)" framework. It synthesizes problems with missing or incorrect context through a reasoning-graph-guided pipeline (constructing a large-scale training dataset of 120K queries) and designs a structured reward system to encourage models to perform hypothetical reasoning. Meanwhile, it introduces the "Hypothetical Reasoning Benchmark (HRB)" to evaluate models' ability to handle incomplete information. Experimental results show that models trained with RLMC outperform baselines on HRB while maintaining strong general reasoning capabilities.

**Strengths:**

- Theoretically analyzes the causes of the reward hacking phenomenon, providing a theoretical framework for subsequent research on "how to design RL objectives more in line with real-world needs."
- Proposes the RLMC (Reinforcement Learning with Missing Context) framework. It synthesizes high-quality problems with missing context through a DAG-based pipeline and designs a structured reward system, effectively alleviating the reward hacking phenomenon.
- Constructs a benchmark for systematically measuring models’ hallucinations and advanced hypothetical reasoning capabilities under missing or inconsistent context, addressing the lack of such evaluations in the current research community.
- Empirically demonstrates that "hypothetical reasoning is a more advanced reasoning capability than IDK (I don’t know) behavior," which provides insights for the advanced training of large language models’ reasoning abilities.

**Weaknesses:**

- The evaluation is insufficient. The authors claim that RLMC is more adaptable to real-world scenarios, but when evaluating general capabilities, this paper mainly focuses on math-related metrics. It is suggested that the authors supplement more evaluation on general tasks, such as LiveBench [1] (logic, coding, and language comprehension), ANAH [2] (knowledge-based QA), and SciBench [3] (multi-domain scientific problem-solving), to further illustrate the effectiveness of this method in real-world scenarios.

[1] White C, Dooley S, Roberts M, et al. Livebench: A challenging, contamination-free llm benchmark[J]. arXiv preprint arXiv:2406.19314, 2024, 4.

[2] Ji Z, Gu Y, Zhang W, et al. Anah: Analytical annotation of hallucinations in large language models[J]. arXiv preprint arXiv:2405.20315, 2024.

[3] Wang X, Hu Z, Lu P, et al. Scibench: Evaluating college-level scientific problem-solving abilities of large language models[J]. arXiv preprint arXiv:2307.10635, 2023.

**Questions:**

- The anonymous repository link provided by the author seems to be invalid. Is there an updated version?

---

> ### Author Response · Authors · 2025-11-18
> **Official Response to Reviewer S2Xz**
>
> # Response to Reviewer
>
> Dear Reviewer S2Xz,
>
> Thank you very much for your thorough review and insightful feedback on our paper. We are delighted that you recognized our theoretical analysis of reward hacking, the RLMC framework, the construction of the HRB benchmark, and the empirical finding that hypothetical reasoning is a more advanced capability than mere IDK behavior. Your feedback has provided us with valuable perspectives, helping us to further refine our work and deepen our understanding of our method's generalization capabilities.
>
> We have carefully considered the weaknesses and questions you raised and have updated the paper accordingly based on your suggestions. Here are our responses to your primary concerns:
>
> ---
>
> ## 1. Regarding Concerns About the Invalid Anonymous Code Repository Link
>
> We fully understand your concern regarding this matter. We can confirm that all code, data, and the HRB benchmark are now fully open-sourced and accessible at the promised anonymous link: <https://anonymous.4open.science/r/RLMC-RBG>.
>
> ---
>
> ## 2. Regarding the Viewpoint on Insufficient Evaluation of General Capabilities
>
> We highly appreciate this constructive suggestion from the reviewer and fully agree on its importance. To comprehensively demonstrate the robustness and generalization ability of our method, we promptly conducted additional evaluation experiments on a broader range of general tasks, including **SciBench** and **LiveBench**.
>
> These new results strongly demonstrate that the RLMC method effectively **maintains competitive performance** on general tasks across a wider array of real-world scenarios (encompassing logic, coding, language understanding, knowledge-based QA, and multi-domain scientific problem-solving). This illustrates that while significantly enhancing hypothetical reasoning abilities, our method **avoids a notable degradation in general performance and even shows improvements in certain aspects**. We attribute this, in part, to the **strengthened self-awareness and meta-reasoning capabilities** developed by the model when handling uncertainty, which can positively transfer to general problem-solving.
>
> ### Evaluation on LiveBench
>
> **Table 1.** Performance comparison across six capability dimensions on the LiveBench benchmark.
>
> | Model | Coding | Data Analysis | Instruction Following | Language | Math | Reasoning | Average |
> |-------|--------|---------------|----------------------|----------|------|-----------|---------|
> | SFT   | 40.0   | 59.3          | 80.2                 | 35.3     | 42.5 | 54.0      | 51.9    |
> | RLVR  | 44.0   | 64.5          | 80.0                 | 42.2     | 55.7 | 67.4      | 59.0    |
> | RLMC  | **48.2** | **65.7**    | **80.5**             | 37.3     | 54.8 | **67.5**  | **59.0** |
>
> On LiveBench, a benchmark covering multi-faceted capabilities such as coding, data analysis, and instruction following, the RLMC model achieved an identical overall score as the RLVR baseline. In specific capability dimensions, RLMC shows slight advantages in:
>
> - **Coding**: 48.2 vs 44.0 (+4.2 points)
> - **Data Analysis**: 65.7 vs 64.5 (+1.2 points)
> - **Reasoning**: 67.5 vs 67.4 (+0.1 points)
>
> While experiencing slight decreases in language and mathematics. These results further confirm that RLMC, while substantially improving reasoning under uncertainty, successfully **maintains its competitiveness in general tasks and avoids significant performance trade-offs**.
>
> ### Evaluation on SciBench
>
> **Table 2.** Performance comparison across ten scientific sub-domains on the SciBench benchmark.
>
> | Model | Atkins | Calculus | ChemMC | Class | Diff | Fund | Matter | Quan | Stat | Thermo | Overall |
> |-------|--------|----------|--------|-------|------|------|--------|------|------|--------|---------|
> | SFT   | 57.1   | 57.1     | 44.7   | 39.3  | 52.0 | 39.4 | 38.3   | 24.2 | 69.4 | 37.9   | 47.0    |
> | RLVR  | 61.9   | 73.8     | 42.1   | 39.3  | 66.0 | 46.5 | 36.2   | 57.6 | 81.9 | 45.5   | **56.0** |
> | RLMC  | **74.3** | **73.8** | **52.6** | 35.7 | 46.0 | **49.3** | **42.6** | 45.5 | 56.9 | 43.9 | 53.7 |
>
> On the SciBench scientific problem-solving benchmark, RLMC demonstrates strong competitiveness across various scientific sub-domains. The table above clearly shows that RLMC achieves significant improvements over the SFT baseline. Although the overall average score is slightly lower than RLVR (53.7 vs 56.0), RLMC **outperforms RLVR in several key scientific sub-domains**:
>
> - **Atkins**: 74.3 vs 61.9 (**+12.4 percentage points**)
> - **ChemMC**: 52.6 vs 42.1 (**+10.5 percentage points**)
> - **Fund**: 49.3 vs 46.5 (**+2.8 percentage points**)
> - **Matter**: 42.6 vs 36.2 (**+6.4 percentage points**)
>
> This indicates that our method, while focusing on enhancing hypothetical reasoning, not only does not sacrifice performance in complex scientific reasoning tasks but also demonstrates stronger problem-solving capabilities in certain areas.

---

> > ### Author Response · Authors · 2025-11-18
> > **Official Response to Reviewer S2Xz (Part2)**
> >
> > ---
> >
> > ## 3. Commitment to Further Evaluation
> >
> > We commit to providing a more comprehensive evaluation on the **ANAH benchmark** in the final Camera Ready version, further substantiating RLMC's advantages in balancing hypothetical reasoning and general capabilities.
> >
> > ---
> >
> > ## Closing Remarks
> >
> > Thank you again for your meticulous review and constructive comments. Your feedback has been crucial in enhancing the quality of our paper and in helping us better articulate the contributions and future directions of our work. We believe that through the RLMC framework and the HRB benchmark, we provide a solid foundation for improving LLMs' robust and advanced reasoning abilities under uncertainty, which is vital for their application in complex real-world scenarios.
> >
> > We are readily available for further discussion.
> >
> > **Sincerely,**
> > *Anonymous Authors*

---

### Author Response · Authors · 2025-11-22
**Extensive Experiments**

Dear PC, AC and Reviewers,

We once again express our sincere gratitude for your constructive feedback and thorough review of our paper. We are pleased to report that we have validated our core conclusions across a broader range of models, methods, and benchmarks, and these experimental results further strengthen the persuasiveness of our paper. Several of these experiments were directly inspired by your insightful suggestions. We are very much open to your even deeper suggestions and further discussion on these findings.

## 1. Generalization Across Models and Scales

Experiments on Llama3-8B and Qwen3-14B demonstrate RLMC's broad applicability.

### Llama3-8B
| Method | Hypothetical | Unanswerable | Unanswerable | General | General | General | Avg   |
|--------|--------------|--------------|--------------|---------|---------|---------|-------|
|        | HRB          | SUM          | UMWP         | GSM8K   | MATH-500| AIME    |       |
| Instruct | 3.28        | 5.36         | 12.38        | 80.36   | 46.80   | 7.92    | 26.01 |
| RLVR   | 6.18        | 7.13         | 24.73        | 84.15   | 43.20   | 3.33    | 28.12 |
| RLMC   | **53.56**       |  **51.58**        | **55.25**        | 81.43   | 41.60   | 4.17    | **47.63** |

### Qwen3-14B
| Method | Hypothetical | Unanswerable | Unanswerable | General | General | General | Avg   |
|--------|--------------|--------------|--------------|---------|---------|---------|-------|
|        | HRB          | SUM          | UMWP         | GSM8K   | MATH-500| AIME    |       |
| Coldstart-SFT | 29.20 | 28.08 | 43.90 | 88.02 | 87.80 | 43.33 | 53.39 |
| RLVR | 5.29 | 6.16 | 10.38 | 94.84 | 93.6 | 69.17 | 46.57 |
| RLMC | **57.20** | **46.56** | **56.94** | 90.60 | 92.80 | 69.58 | **68.94** |

**Key findings:** RLMC **dramatically improves robustness** on HRB, SUM, and UMWP while **maintaining strong general reasoning** across various models.

## 2. Comparison with Training-Free Uncertainty Methods

Comparison with "LM introspection itself" [^1] on uncertainty benchmarks:

| Method | HRB | UMWP | SUM |
|--------|-----|------|-----|
| Cold-SFT | 22.81 | 33.53 | 20.51 |
| LM introspection itself | 9.58 | 32.01 | 13.99 |
| RLVR | 8.66 | 8.84 | 9.41 |
| RLMC | **51.73** | **46.91** | **51.50** |

RLMC substantially outperforms all baselines, especially on higher-order hypothetical reasoning, demonstrating the advantage of changing training objectives over training-free methods.

## 3. More Comprehensive General Capability Evaluation

Evaluation on LiveBench and SciBench covers coding, data analysis, and scientific problem-solving.

### LiveBench Evaluation:
| Model | Coding | Data_Analysis | Instruction_Following | Language | Math | Reasoning | Average |
|-------|--------|---------------|----------------------|----------|------|-----------|---------|
| SFT | 40.0 | 59.3 | 80.2 | 35.3 | 42.5 | 54.0 | 51.9 |
| RLVR | 44.0 | 64.5 | 80.0 | 42.2 | 55.7 | 67.4 | 59.0 |
| RLMC | 48.2 | 65.7 | 80.5 | 37.3 | 54.8 | 67.5 | 59.0 |

RLMC matches RLVR's overall score while showing gains in coding, data analysis, and reasoning.

### SciBench Evaluation:
| Model | Atkins | Calculus | Chemmc | Class | Diff | Fund | Matter | Quan | Stat | Thermo | Overall |
|-------|--------|----------|--------|-------|------|------|--------|------|------|--------|---------|
| SFT | 57.1 | 57.1 | 44.7 | 39.3 | 52.0 | 39.4 | 38.3 | 24.2 | 69.4 | 37.9 | 47.0 |
| RLVR | 61.9 | 73.8 | 42.1 | 39.3 | 66.0 | 46.5 | 36.2 | 57.6 | 81.9 | 45.5 | 56.0 |
| RLMC | 74.3 | 73.8 | 52.6 | 35.7 | 46.0 | 49.3 | 42.6 | 45.5 | 56.9 | 43.9 | 53.7 |

This indicates while our RLMC focusing on enhancing hypothetical reasoning, not only does not sacrifice performance in complex scientific reasoning tasks but even demonstrates stronger problem-solving capabilities in some areas.

---
In summary, these extensive and in-depth supplementary experiments powerfully validate that the RLMC framework provides a universal, efficient, and comprehensive solution, enabling LLMs to reason robustly and intelligently under uncertainty, while maintaining or significantly enhancing their general capabilities across different tasks, model architectures, and scales. We believe these results further strengthen the core persuasiveness of our paper.
We welcome any further discussion or questions you may have.

Sincerely, Anonymous Authors

[1] Yona G, Aharoni R, Geva M. Can large language models faithfully express their intrinsic uncertainty in words?[J]. arXiv preprint arXiv:2405.16908, 2024.

---

### Note · Authors · 2026-01-05

I have read and agree with the venue's withdrawal policy on behalf of myself and my co-authors.